# Mitigating Dataset Harms Requires Stewardship: Lessons from 1000 Papers

**Kenny Peng, Arunesh Mathur, Arvind Narayanan**
Princeton University

## Abstract

Machine learning datasets have elicited concerns about privacy, bias, and unethical applications, leading to the retraction of prominent datasets such as DukeMTMC, MS-Celeb-1M, and Tiny Images. In response, the machine learning community has called for higher ethical standards in dataset creation. To help inform these efforts, we studied three influential but ethically problematic face and person recognition datasets—Labeled Faces in the Wild (LFW), MS-Celeb-1M, and DukeMTMC— by analyzing nearly 1000 papers that cite them. We found that the creation of derivative datasets and models, broader technological and social change, the lack of clarity of licenses, and dataset management practices can introduce a wide range of ethical concerns. We conclude by suggesting a distributed approach to harm mitigation that considers the entire life cycle of a dataset.

## 1 Introduction

Datasets play an essential role in machine learning research but also raise ethical concerns. These concerns include the privacy of individuals included [45, 70], representational harms introduced by annotations [25, 44], effects of biases on downstream use [20, 21, 18], and use for ethically dubious purposes [45, 75, 66]. These concerns have led to the retractions of prominent research datasets including Tiny Images [81], VGGFace2 [68], DukeMTMC [72], and MS-Celeb-1M [41].

The machine learning community has responded to these concerns and has developed ways to mitigate harms associated with datasets. Researchers have worked to make sense of ethical considerations involved in dataset creation [43, 69, 32], have proposed ways to identify and mitigate biases in datasets [11, 82], have developed means to protect the privacy of individuals in datasets [70, 91], and have improved methods to document datasets [35, 47, 12, 65].

The premise of our work is that these efforts can be more effective if informed by an understanding of how datasets are used in practice. We present an account of the life cycles of three popular face and person recognition datasets: Labeled Faces in the Wild (LFW) [49], MS-Celeb-1M [41], and DukeMTMC [72]. These datasets have been the subject of recent ethical scrutiny [43] and, in the case of MS-Celeb-1M and DukeMTMC, have been retracted by their creators. Analyzing nearly 1,000 papers that cite these datasets and their derivative datasets or pre-trained models, we present five findings that describe ethical considerations arising beyond dataset creation:

- Dataset retraction has a limited effect on mitigating harms (Section 3). Our analysis shows that even after DukeMTMC and MS-Celeb-1M were retracted, their underlying data remained widely available and continued to be used in research papers. Because of such "runaway data," retractions are unlikely to cut off data access; moreover, without a clear indication of the underlying intention, retractions may have limited normative influence.

- Derivatives raise new ethical concerns (Section 4). The derivatives of DukeMTMC, MS-Celeb-1M, and LFW that we document enable the use of the dataset in production settings, introduce new

Table 1: A summary of our overarching analysis of MS-Celeb-1M, DukeMTMC, and LFW.

|  | MS-Celeb-1M | DukeMTMC | LFW |
|---|---|---|---|
| **Papers that cite dataset or derivatives** | 1,404 | 1,393 | 7,732 |
| **Papers sampled for analysis** | 276 | 275 | 400 |
| **Papers in sample that use dataset or derivatives** | 179 | 114 | 152 |
| **Derivative datasets identified** | 8 | 7 | 20 |
| **Pre-trained models identified** | 21 repositories | 0 | 0 |

annotations of the data, or apply additional data processing steps. Each of these alterations lead to a unique set of ethical considerations.

- Licenses, a primary mechanism governing dataset use, can lack substantive effect (Section 5). We found that the licenses of DukeMTMC, MS-Celeb-1M, and LFW do not effectively restrict production use of the datasets. In particular, while the original license of MS-Celeb-1M only permits non-commercial research use of the dataset, only 3 of 21 GitHub repositories we found containing models pre-trained on MS-Celeb-1M included the same designation. We found anecdotal evidence suggesting that production use of models trained on non-commercial datasets is commonplace.

- The ethical concerns associated with a dataset can change over time, as a result of both technological and social change (Section 6). In the case of LFW and the influential ImageNet dataset [28], technological advances opened the door for production use of the datasets, raising new ethical concerns. Additionally, various social factors led to a more critical understanding of the demographic composition of LFW and the annotation practices underlying ImageNet.

- While dataset management and citation practices can support harm mitigation, current practices have several shortcomings (Section 7). Dataset documentation is not easily accessible from citations and is not persistent. Moreover, dataset use is not clearly specified in academic papers, often resulting in ambiguities. Finally, current infrastructure does not support the tracking of dataset use or of derivatives in order to retrospectively understand the impact of datasets.

Based on these findings, we revisit existing recommendations for mitigating the harms that arise from datasets, and adapt them to encompass the broader set of concerns we describe here. Our approach emphasizes steps that can be taken after dataset creation, which we call dataset stewarding. We advocate for responsibility to be distributed among many stakeholders including dataset creators, conference program committees, dataset users, and the broader research community.

## 2 Overview of datasets and analysis

We first collected a list of 54 face and person recognition datasets (listed in Appendix B), and chose three popular ones for a detailed analysis of their life cycles: Labeled Faces in the Wild (LFW) [49], DukeMTMC [72], and MS-Celeb-1M [41]. We chose LFW because it was the most cited in our list and allows for longitudinal analysis since it was introduced in 2007.[1] We chose DukeMTMC and MS-Celeb-1M because they were the most cited datasets in our list that had been retracted. We refer to these three datasets as *parent datasets*. We describe them in detail in Appendix C.

We began our analysis by constructing a corpus of papers that cited—and potentially used—each parent dataset or its derivatives (we use the term *derivative* broadly, including datasets that contain the original images, datasets that provide additional annotations, as well as models pre-trained on the dataset). To do this, we first compiled a list of derivatives of each parent dataset and associated them with their research papers. We then compiled a list of papers citing each of these associated papers using the Semantic Scholar API [34]. The first author coded a sample of these papers, recording whether a paper used the parent dataset or a derivative as well as the name of the parent dataset or derivative. In total, our analysis included 946 unique papers, including 275 citing DukeMTMC or its derivatives, 276 citing MS-Celeb-1M or its derivatives, and 400 citing LFW or its derivatives. We found many papers using derivatives that were not included in our original list of derivatives, which we consider an unavoidable limitation since we are not aware of a systematic way to find all

---

[1]LFW had slightly fewer total citations than one other dataset in our list, Yale Face Database B [36], but LFW has been cited significantly more times per year, especially in recent years.

Table 2: A summary of the status of MS-Celeb-1M and DukeMTMC after their April 2019 retractions.

| | MS-Celeb-1M | DukeMTMC |
|---|---|---|
| Availability of original | The dataset is still available through Academic Torrents and Archive.org. | We did not find any locations where the original dataset is still available. |
| Availability of derived datasets | We found five derived datasets that remain available with images from the original. | We found two derived datasets that remain available with images from the original. |
| Availability of pre-trained models | We found 20 GitHub repositories containing models pre-trained on MS-Celeb-1M that remain available. | We did not find any models pre-trained on DukeMTMC data that are still available. |
| Continued use | In our 20% sample, MS-Celeb-1M and its derivatives were used 54 times in papers published in 2020. | In our 20% sample, DukeMTMC and its derivatives were used 73 times in papers published in 2020. |
| Status of original dataset page | Website (`https://www.msceleb.org`) only contains filler text. | Website (`http://vision.cs.duke.edu/DukeMTMC/`) returns a DNS error. |
| Other statements made by creators | In June 2019, Microsoft said in response to a press inquiry that the dataset was taken down "because the research challenge is over" [66]. | A creator of DukeMTMC apologized in June 2019, noting that they had violated IRB guidelines [80], but this explanation did not appear in official channels. |
| Availability of metadata | The license is no longer officially available. It was previously available through the website, which was taken down in April 2019. Notably, the license prohibits distribution of the dataset or derivatives. | The license is no longer officially available, but is still available through GitHub repositories of derivative datasets. |

derivatives. Because our corpus does not contain *all* papers using the parent dataset or a derivative, our results should be viewed as lower bounds throughout. We further note that most of our analyses do not address use outside published research. We provide additional details about our methods in Appendix D.

## 3 Retractions and runaway data

When datasets are deemed problematic by the machine learning community, activists, or the media, dataset creators have responded by retracting them. MS-Celeb-1M [41], DukeMTMC [72], VG-GFace2 [23], and Brainwash [78] were all retracted after an investigation by Harvey and Laplace [45] highlighted ethical concerns with how the data was collected by the creators and being used by the community. TinyImages [81] was retracted after Prabhu and Birhane [70] raised ethical concerns about offensive labels and a lack of consent by data subjects.

Retractions such as these may mitigate harm in two primary ways. First, they may place hard limitations on dataset use by making the data unavailable. Second, they may exert a normative influence, indicating to the community that the data should no longer be used. This can allow publication venues and other bodies to place their own limitations on such use.

With this in mind, we analyzed the retractions of MS-Celeb-1M [2] and DukeMTMC, summarized in Table 2. We find that both retractions fall short of effectively accomplishing either of the above mentioned goals. Since the underlying data was available through many different sources (i.e., the data had "runaway" [45]), both datasets remain available despite the retraction of the parent dataset. And because the dataset creators did not clearly state that the datasets should no longer be used, they may have left users confused, contributing to their continued use (see Figure 1).

---

[2]Although the creators of MS-Celeb-1M never officially stated that the dataset was removed due to ethical concerns, we consider its removal a *retraction* in the sense that its removal responded to ethical concerns. The removal followed mere days after the second of two critical reports of the dataset [45, 66]. Furthermore, the reason Microsoft gave for the removal—that "the research challenge is over" [66]—is not, by itself, a common reason to remove a dataset.

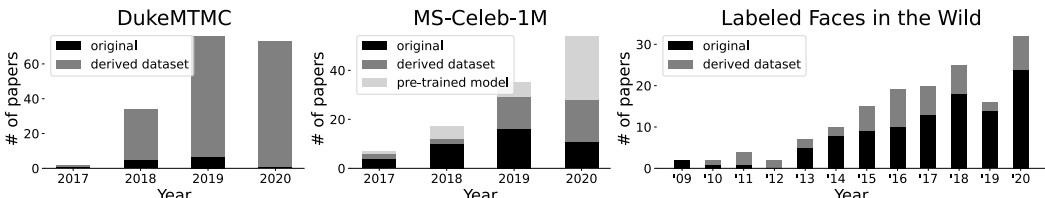

Figure 1: The use of DukeMTMC, MS-Celeb-1M, LFW, and their derivatives over time. All three datasets were commonly used through derivatives. DukeMTMC and MS-Celeb-1M were retracted in April 2019, but continued to be used in 2020—largely, through derivatives.

There are many similarities between the continued use of retracted datasets and the continued citation of retracted papers, which is a well-known yet persistent challenge [26]. Several studies have shown that articles continue to be cited after retraction (e.g., [22, 9, 74]). One reason might be because the retraction status is often not clear in all locations where a paper is available [26]. Two primary types of interventions have been proposed to limit continued citation. The first involves making the retraction status of articles more clear and accessible [19, 67]. The second involves publication venues requiring authors to check that their reference list includes no retracted papers [76, 9]. The same types of interventions are applicable in the case of retracted datasets, and are reflected in the recommendations we provide in Section 8.

In addition to questions of efficacy, retraction can come into tension with efforts to archive datasets. In work critiquing machine learning datasets, Crawford and Paglen [25] note the issue of "inaccessible or disappearing datasets," writing that "If they are, or were, being used in systems that play a role in everyday life, it is important to be able to study and understand the worldview they normalize. Developing frameworks within which future researchers can access these data sets in ways that don't perpetuate harm is a topic for further work."

## 4 Derivatives raise new ethical questions

Machine learning datasets often serve simultaneous roles as a specific tool (e.g., a benchmark for a particular task) and as a collection of raw material that may be leveraged for other purposes. Derivative creation falls into the latter category, and can be seen as a success of resource-sharing in the machine learning community as it reduces the cost of obtaining data. This also means that the effort or cost of creating an ethically-dubious derivative can be much less than creating a similar dataset from scratch. For example, the DukeMTMC-ReID dataset was created using annotations and bounding boxes from the original dataset to build a cropped subset for benchmarking person re-identification. This process can be entirely automated (as far as we can determine from available documentation), which is far cheaper and faster than collecting and manually annotating videos.

In our analysis, we identified four ways in which a derivative can raise ethical considerations (which does not necessarily imply that the creation of the derivative or the parent dataset is unethical). We analyzed all the 41 derivatives of MS-Celeb-1M, DukeMTMC, and LFW based on the four categories we identified. The full matrix is in Table 4 in the appendix; we summarize the four categories below.

**New application.**   Either implicitly or explicitly, modifications of a dataset can enable applications raising new ethical concerns. Twenty-one of 41 derivatives we identified fall under this category. For example, DukeMTMC-ReID, a person re-identification benchmark, is used much more frequently than DukeMTMC, a multi-target multi-camera tracking benchmark. While these problems are similar, they may have different motivating applications. SMFRD [92] is a derivative of LFW that adds face masks to its images. It is motivated by face recognition applications during the COVID-19 pandemic, when many people wear face-covering masks. "Masked face recognition" has been criticized for violating the privacy of those who may want to conceal their face (e.g., [63, 90]).

**Pre-trained models.**   We found six model classes that were commonly trained on MS-Celeb-1M. Across these six classes, we found 21 GitHub repositories that released models pre-trained on MS-Celeb-1M. These pre-trained models can be used out-of-the-box to perform face recognition or can be used for transfer learning. This enables the use of MS-Celeb-1M for a wide range of applications,

albeit in a more indirect way. There are also concerns about the effect of biases in training data on pre-trained models and their downstream applications [77].

**New annotations.**   The annotation of data can also result in privacy and representational harms. (See Section 3.1 of [69] for a survey of work discussing representational concerns.)  Seven of 41 derivatives fall under this category. Among the derivatives we examined, four annotated the data with gender, three with race or ethnicity, and two with additional attributes such as "attractiveness." Such annotations may also enable research in ethically dubious applications such as the classification and identification of people via sensitive attributes.

**Other post-processing.**   Other derivatives neither repurpose the data for new applications nor contribute annotations. Rather, these derivatives are designed to aid the original task with more subtle modifications. Still, even minor modifications can raise ethical questions. Five of 41 derivatives (each of MS-Celeb-1M) "clean" the original dataset, creating a more accurate set of images from the original, which is known to be noisy. This process often reduces the number of images significantly, after which, we may be interested in the resulting composition. For example, does the cleaning process reduce the number of images of people of a particular demographic group? Such a shift may impact the downstream performance of such a dataset. Five of 41 derivatives (each of LFW) align, crop, or frontalize images in the original dataset. Here, too, we may ask about how such techniques perform on different demographic groups.

## 5   Effectiveness of licenses

Licenses, or terms of use, are legal agreements between the creator and users of datasets, and often dictate how the dataset may be used, derived from, and distributed. We focus on the role of a license in harm mitigation, i.e., as a tool to restrict unintended and potentially harmful uses of a dataset.

By analyzing the licenses of DukeMTMC, MS-Celeb-1M, LFW, and ImageNet, and whether restrictions were inherited by derivatives, we found several shortcomings of licenses as a tool for mitigating harms through preventing commercial use. We included ImageNet in this analysis because we discovered in preliminary research that there is confusion around the implications of ImageNet's license (which allows only non-commercial research use) on pre-trained models. Our findings are summarized in Table 3.

Motivated by these findings, we further sought to understand whether models trained on datasets released for non-commercial research are being used commercially. Such use can exacerbate the real-world harm caused by datasets. Due to the obvious difficulties involved in studying this question, we approach it by studying online discussions. We identified 14 unique posts on common discussion sites that inquired about the legality of using pre-trained models that were trained on non-commercial datasets.

From these posts, we found anecdotal evidence that non-commercial dataset licenses are sometimes ignored in practice.  One response reads: "More or less everyone (individuals, companies, etc) operates under the assumption that licences on the use of data do not apply to models trained on that data, because it would be extremely inconvenient if they did." Another response reads: "I don't know how legal it really is, but I'm pretty sure that a lot of people develop algorithms that are based on a pretraining on ImageNet and release/sell the models without caring about legal issues. It's not that easy to prove that a production model has been pretrained on ImageNet ..." Commonly-used computer vision frameworks like Keras and PyTorch include models pre-trained on ImageNet, making the barrier for commercial use low.

In responses to these posts, representatives of Keras and PyTorch suggested that such use is generally allowed, but that they could not provide an official answer. The representative for PyTorch wrote that according to their legal team's guidance, "weights of a model trained on that data may be considered derivative enough to be ok for commercial use. Again, this is a subjective matter of comfort. There is no publishable 'answer' we can give." The representative for Keras wrote that "In the general case, pre-trained weight checkpoints have their own license which isn't inherited from the license of the dataset they were trained on. This is not legal advice, and you should consult with a lawyer."

While we don't comment on the legality of these practices, we note that they represent a potential legal loophole. If a company were to train a model on ImageNet for commercial purposes, it would

Table 3: Dataset creators may intend to prohibit commercial use, but licenses do not effectively accomplish this.

| Dataset | Non-commercial intention | License shortcomings | Evidence of comm. use |
|---|---|---|---|
| MS-Celeb-1M | Users may "use and modify this Corpus for the limited purpose of conducting non-commercial research." | Implication on pre-trained models is unclear. The license is no longer publicly available. | We found 18 GitHub repositories containing models pre-trained on MS-Celeb-1M data and released under commercial licenses. |
| LFW | "... it should not be used to conclude that an algorithm is suitable for any commercial purpose." | No license was issued. A disclaimer was added in 2019 (excerpted on left), but carries no legal weight. | We identified four commercial systems that actively advertise their performance on LFW. |
| ImageNet | "Researcher shall use the Database only for non-commercial research and educational purposes." | The license does not prevent re-distribution of the data or pre-trained models under commercial licenses. | We found nine GitHub repositories containing models pre-trained on ImageNet and released under commercial licenses. Keras, PyTorch, and MXNet include pre-trained weights. |
| DukeMTMC | "You may not use the material for commercial purposes." | Implication on pre-trained models is unclear. Government use is not "commercial," but can raise similar or greater ethical concerns. | We did not find clear evidence suggesting commercial use of DukeMTMC. |

be a relatively clear license violation; yet, the practice of downloading pre-trained models, which has substantively the same effect, appears to be common. Similarly, derivatives that don't inherit the license restrictions of the original dataset may also represent a loophole. Dataset creators can avoid such unintended uses by being much more specific in their licenses. For example, The Montreal Data License [13] allows for dataset creators to specify restrictions to models trained on the dataset.

We caution that our analysis in this section is preliminary and that the evidence we have presented is tentative and anecdotal. A more thorough study could be conducted through interviews or surveys of practitioners to further illuminate their common practices, legal understanding, as well as the extent to which legal understanding shapes practice.

## 6  Technological and social change affects dataset ethics

We now examine how ethical considerations associated with a dataset change over time. For this analysis, we used LFW and ImageNet, but not DukeMTMC and MS-Celeb-1M as they are relatively recent and thus less fit for longitudinal analysis. We observed that ethical concerns involving both datasets surfaced more than a decade after release. We identified several factors—including increasing production viability, evolving ethical standards, and changing academic incentives—that may help explain this delay.

**Changing ethics of LFW.**  LFW was introduced in 2007 to benchmark face verification. It is considered the first "in-the-wild" face recognition benchmark, designed to help face recognition improve in unconstrained settings. The production use of the dataset was unviable in its early years, one indication being that the benchmark performance on the dataset was poor.[3] Over time, LFW became a standard benchmark and technology improved. This opened to door for increased use of LFW to benchmark commercial systems, as illustrated in Figure 2.

This type of use inspired ethical concerns, as benchmarking production systems has greater real-world potential for harm than benchmarking models used for research. The production use of facial

---

[3]Consider the benchmark task of face verification with unrestricted outside training data (the easiest of the tasks proposed in [49]). The best reported accuracy as of 2010 was only 84.5%, whereas today it is 99.9%.

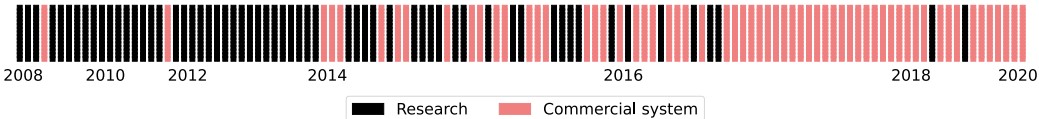

Figure 2: A visualization of the rise of the production use of LFW, based on data from LFW's website. By examining versions of the website archived on the Wayback Machine, we identified (approximately) the year in which different results were added. Only 3 of 38 results added before 2014 were commercial but 41 of 49 results after 2016 were commercial.

recognition systems in applications such as surveillance or policing have caused backlash—especially because of disparate performance on minority groups.

In 2019—more than a decade after the dataset's release—a disclaimer was added to the dataset's website noting that it should not be used to verify the performance of commercial systems [2]. Notably, this disclaimer emphasized LFW's insufficient diversity across many demographic groups, as well as in pose, lighting, occlusion, and resolution. In contrast, when the dataset was first released, the creators highlighted the dataset's diversity: LFW contained real-world images of people, whereas past datasets had mostly contained images taken in a laboratory setting [71]. This shift may be partially due to recent work showing disparate performance of face recognition on different demographic groups and highlighting the need for demographically-diverse benchmarks [20].

**Changing ethics of ImageNet.** When ImageNet was introduced, object classification was still immature. Today, as real-world use of such technology has become widespread, ImageNet has become a common source for pre-training, again illustrating the shift from research to production use. As discussed in Section 5, even as the dataset's terms of service specify non-commercial use, the dataset is commonly used in pre-trained models released under commercial licenses.

We also consider how social factors have shaped recent ethical concerns. In 2019, researchers revealed that many of the images in the "people" category of the dataset were labeled with misogynistic and racial slurs and perpetuated stereotypes, after which images in these categories were removed [25, 70]. This work critiquing ImageNet first appeared nearly a decade after its release (even if issues were known to some earlier). As it is reasonable to assume that the labels used in ImageNet would have been considered offensive in 2009, the lag between the dataset's release and the removal of such labels is noteworthy. We propose three factors that have changed since the release of ImageNet and hypothesize that they may account for the lag. First, public concern over machine learning datasets and applications has grown. Issues involving datasets have received significant public attention—the article by Crawford and Paglen [25] accompanied several art exhibitions and the topic has been covered by many media outlets (e.g., [66, 62, 75]). Relatedly, academic incentives have changed and critical work is more easily publishable. Related work highlighting assumptions underlying classification schemes [14, 52] have been published in FAccT, a conference focused on fairness, accountability, and transparency in socio-technical systems that was only founded in 2018. Finally, norms regarding the ethical responsibility of dataset creators and machine learning researchers more generally have shifted. These norms are still evolving; responses to recently-introduced ethics-related components of peer review have been mixed [7].

The transition from research to production use, in some sense, is a sign of success of the dataset, and thus may be anticipated. Benchmark datasets in machine learning are typically introduced for problems that are not yet viable in production use cases; and should the benchmark be successful, it will help lead to the realization of real-world application. The ethics of LFW and ImageNet were also each shaped by social factors, if in different ways. Whereas shifting ethical standards contributed to changing views of LFW, ImageNet labels would likely have been considered offensive when the dataset was first created. For ImageNet, social factors seem to have led to evolving incentives to identify and address ethical issues. While "what society deems fair and ethical changes over time" [17], additional factors can dictate if and how these standards are operationalized.

# 7 Dataset management and citation

We turn to the role of dataset management and citation in harm mitigation. By dataset management, we mean storing a dataset and associated metadata. By dataset citation, we mean the referencing of a dataset used in research with the aim of facilitating access to the dataset and metadata. We give three reasons for why dataset management and citation are important for mitigating harms caused by datasets: facilitating documentation accessibility, transparency and accountability, and tracking of dataset use. We then summarize how current practices fall short in achieving these aims.

**Documentation.**  Access to dataset documentation facilitates responsible dataset use. Documentation can provide information about a dataset's composition, its intended use, and any restrictions on its use (through licensing information, for example). Many researchers have proposed documentation tools for machine learning datasets with harm mitigation in mind [35, 12]. Dataset management and citation can ensure that documentation is easily accessible, even if the dataset itself is not or is no longer publicly accessible. In Section 3 and Section 5, we discussed how retracted datasets no longer included key information such as licensing information, potentially leading to confusion. For example, with MS-Celeb-1M's license no longer publicly available, the license status of derivative datasets, pre-trained models, and remaining copies of the original is unclear.

**Transparency and accountability.**  Dataset citation facilitates transparency in dataset use, in turn facilitating accountability. By clearly indicating the dataset used and where information about the dataset can be found, researchers become accountable for ensuring the quality of the data and its proper use. Different stakeholders, such as the dataset creator, program committees, and other actors can then hold researchers accountable. For example, if proper citation practices are followed, peer reviewers can more easily check whether researchers complied with dataset licenses.

**Tracking.**  Large-scale analysis of dataset use—as we do in this paper—can illuminate a dataset's impact and potential avenues of risk or misuse. This knowledge can allow dataset creators to update documentation, better establishing intended use. Citation infrastructure supports this task by collecting such use in an organized manner. This includes both tracking the direct use of a dataset in academic research, as well as the creation of derivatives.

Our findings, summarized below, suggest that current dataset management and citation practices fall short in supporting the above goals. A complete set of findings is given in Appendix H.

- **Datasets and metadata are not persistent.** None of the 38 datasets in our analysis are managed through shared repositories, a common practice in other scientific fields. We were unable to locate three datasets, and two more are only available through the Wayback Machine. After DukeMTMC and MS-Celeb-1M's retractions, their licenses are no longer officially available.

- **Disambiguating citations is hard.** None of the 38 datasets have DOIs or stable identifiers. Only six of 60 sampled papers provided access information such as a URL. We encountered difficulties accessing datasets when no URL is given, as five of the datasets did not even have names. The current practice of citing datasets via a combination of name, description, and associated papers makes even manual disambiguation challenging. We were unable to disambiguate a citation in 42 of 446 cases and encountered difficulties in roughly 50 additional cases.

- **Tracking is difficult.** The lack of dataset-specific identifiers makes systematic tracking hard. Papers using a dataset may not cite a particular paper, and vice versa. Moreover, there is no way to systematically identify the derivatives of a dataset.

# 8 Recommendations

In the last few years, there have been numerous recommendations for mitigating the harms associated with machine learning datasets. Researchers have proposed frameworks for dataset and model documentation [35, 12, 47], which can both guide responsible dataset creation and facilitate responsible use. Other researchers have proposed guidelines for ethical data collection [43], drawing from "interventionist" approaches modeled on archives [51] or focusing on specific principles, such as requiring informed consent from data subjects [70]. Still others have created tools for identifying and

mitigating biases [11, 82] or preserving privacy [70, 91] in datasets. Our own recommendations build on this body of work and aren't meant to replace existing proposals.

That said, previous approaches primarily consider dataset creation. As we have shown, ethical impacts are hard to anticipate and address at dataset creation time. Thus, we argue that harm mitigation requires stewarding throughout a dataset's life cycle. Our recommendations reflect this understanding.

We contend that the problem cannot be left to any one stakeholder such as dataset creators or IRBs. We propose a more distributed approach where many stakeholders share responsibility for ensuring the ethical use of datasets. We assume the willingness of dataset creators, program committees, and the broader research community; addressing harms from callous or malicious users or outside the research context is beyond our scope. Below, we present recommendations for dataset creators, conference program committees, dataset users, and other researchers. In Appendix J, we discuss how IRBs—which hold traditional oversight over research ethics—are an imperfect fit for dataset-centered research and should not be relied on for regulating machine learning datasets or their use.

Our recommendations are informed by the principle of separating blame from responsibility. Even if an entity is not to blame for a particular harm, that entity might be well positioned to reduce the likelihood of that harm occurring. For example, as a response to ML research that develops technologies that could be used used to violate human rights, it is reasonable to allocate some responsibility to conference program committees to prohibit this type of research. Similarly, as a response to harms associated with data, it is reasonable to allocate some responsibility to dataset creators. As we argue below, there are many ways in which dataset creators can minimize the chances of downstream abuse.

## 8.1 Dataset creators

We make two main recommendations for dataset creators, both based on the normative influence they can exert and based on the harder constraints they can impose on how datasets are used.

**Make ethically salient information clear and accessible.** Dataset creators can place restrictions on dataset use through licenses and provide other ethically salient information through other documentation. But in order for these restrictions to be effective, they must be clear. In our analysis, we found that licenses are often insufficiently specific. For example, when restricting the use of a dataset to "non-commercial research," creators should be explicit about whether this also applies to models trained on the dataset. It may also be helpful to explicitly prohibit specific ethically questionable uses.

In order for licenses or documentation to be effective, they also need to be accessible. Licenses and documentation should be persistent, which can be accomplished through the use of standard data repositories. Dataset creators should also set requirements for dataset users and creators of derivatives to ensure that this information is easy to find from citing papers and derived datasets.

These recommendations also apply to dataset retraction. Retractions should be explicit and easily accessible. Moreover, dataset creators should seek to make the retraction status visible wherever the dataset or its derivatives remain available.

**Actively steward the dataset and exert control over use.** Throughout our analysis, we show how ethical considerations can evolve over time. Dataset creators should continuously steward a dataset, actively examining how it may be misused, and making updates to the license, documentation, or access restrictions as necessary. A minimal access restriction is for users to agree to terms of use. A more heavyweight process in which dataset creators make case-by-case decisions about access requests can be used in cases of greater risk. The Fragile Families Challenge is an example of this [60].

Based on our analysis in Section 3 and Section 4, derivative creation often raises ethical risks. We showed that derivatives can make data more widely available—in many cases, without the original licensing information. Additionally, derivatives may introduce new ethical concerns, such as through enabling new applications. A dataset's terms of use can establish guidance for derivative creation. This may include a list of specifically allowed (or disallowed) types of derivatives, in addition to distribution and licensing requirements. Of course, dataset creators may not be able to anticipate all potential ethically-dubious derivatives in advance. Creators may overcome this by requiring explicit permission be obtained unless the derivative belongs to a pre-approved category.

We recognize that dataset stewarding increases the burden on dataset creators. In our discussions with dataset creators, we heard that creating datasets is already an undervalued activity and that a norm of dataset stewardship might further erode the incentives for creators. We acknowledge this concern, but maintain that there is an inherent tension between ethical responsibility and minimizing burdens on dataset creators. One solution is for dataset creation to be better rewarded in the research community; some of our suggestions for program committees below may have this effect.

## 8.2   Conference program committees

**Use ethics review to encourage responsible dataset use.**   PCs are in a position to govern both the creation of datasets (and derivatives) and the use of datasets through ethics reviews of the associated papers. PCs should develop clear guidelines for ethics review. For example, PCs can require researchers to clearly state the datasets used, justify the reasons for choosing those datasets, and certify that they complied with the terms of use of each dataset. In particular, PCs can require researchers to examine if a dataset has been retracted. Some conferences, such as NeurIPS, already have ethics guidelines relating to dataset use.

**Encourage standard dataset management and citation practices.**   PCs should consider standardized dataset management and citation requirements, such as requiring dataset creators to upload their dataset and supporting documentation to a public data repository. Guidelines on effective dataset management and citation practices can be found in [85]. The role of PCs is particularly important for dataset management and citation, as these practices benefit from community-wide adoption.

**Introduce a dataset-specific track.**   NeurIPS now includes a track specifically for datasets. The introduction of such tracks facilitates more careful and tailored ethics reviews for datasets. The journal *Scientific Data* is devoted entirely to describing datasets.

**Allow advance review of datasets and publications.**   We tentatively recommend that conferences can allow researchers to submit proposals for datasets prior to creation. By receiving preliminary feedback, researchers can be more confident that their dataset both will be valued and will pass initial ethics reviews. This mirrors the concept of "registered reports" [6], in which a proposed study is peer reviewed before it is undertaken and provisionally accepted for publication *before* the outcomes are known, as a way to counter publication biases.

## 8.3   Dataset users and other researchers

At a minimum, dataset users should comply with the terms of use of datasets. But their responsibility goes beyond compliance. They should also carefully study the accompanying documentation and analyze the appropriateness of using the dataset for their particular application (e.g., whether dataset biases may propagate to models). Dataset users should also clearly indicate what dataset is being used in their research papers and ensure that readers can access the dataset based on the information provided. As we showed in Section 7, traditional paper citations often lead to ambiguity.

We showed how a dataset's impact is not fully understood at the time of its creation. We recommend that the community systematize the retrospective study of datasets to understand their shortcomings and misuse. Researchers should not wait until the problems become serious and there is backlash.

It is especially important to understand how datasets and pre-trained models are being used in production settings, which our work does not address. Policy interventions may be necessary to incentivize companies to be transparent about datasets or models used in deployed pipelines.

## 9   Conclusion

The machine learning community is responding to a wide range of ethical concerns regarding datasets and asking fundamental questions about the role of datasets in machine learning research. In this paper, we provided a new perspective. Through our analysis of the life cycles of three datasets, we showed how developments that occur after dataset creation can impact the ethical consequences, making them hard to anticipate a priori. We advocated for an approach to harm mitigation in which responsibility is distributed among stakeholders and continues throughout the life cycle of a dataset.

## Acknowledgments

This work is supported by the National Science Foundation under Awards IIS-1763642 and CHS-1704444. We thank participants of the Responsible Computer Vision (RCV) workshop at CVPR 2021, the Princeton Bias-in-AI reading group, and the Princeton CITP Work-in-Progress reading group for useful discussion. We also thank Solon Barocas, Marshini Chetty, Sayash Kapoor, Mihir Kshirsagar, and Olga Russakovsky for their helpful feedback.

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
