# OpenReview forum: "Mitigating dataset harms requires stewardship: Lessons from 1000 papers"
_NeurIPS.cc/2021/Track/Datasets_and_Benchmarks/Round2 — NeurIPS 2021 Datasets and Benchmarks Track (Round 2)_

### Official Review · Reviewer_83pV · 2021-09-19
**Review for Paper**

**Rating:** 8
**Confidence:** 5
**Correctness:** Yes, from what I can tell.
**Clarity:** Yes.

**Strengths:**

- Novel idea (although the retraction of the datasets shows that the problem itself is known, but the idea to empirically study and mitigate the problem is interesting)
- Really enjoyed the "changing ethics" section. I think it is a very important point that we often don't highlight enough and I am glad the authors discussed it in detail.
- Figure 2 is especially illuminating. Clever use of graphics to highlight the point of harms "leaking" from academic papers into the industry over time.
- Especially glad for the summary tables for the three datasets discussed. It is a great reference point to look up and for a paper with no actual experiments, the table really bolsters the need for the existence of the paper.

**Weaknesses:**

- Although I really enjoyed the legal loopholes part, I am not sure if looking at online discussions is enough to validate license confusion (sampling bias)? A more principled survey might have been more scientifically valid, especially given that FairML is a more interdisciplinary area than other areas of CS. There might be folks in non-STEM disciplines that react differently to licenses (by virtue of being more knowledgeable about the law, for instance law or policy researchers).

**Additional Feedback:**

-

**Documentation:**

Yes. The authors provide copious details in the supplementary materials section. They also provide a repo link with the csvs of their observations of the papers from Semantic Scholar.

**Ethics:**

No, the paper seems ethically valid and justified. There is no new human subject research.

**Relation To Prior Work:**

Yes.

**Summary And Contributions:**

The paper highlights the temporal harms of a dataset -- harms that might not have been widely known at the time of release but became clearer as the dataset proliferated into academic and commercial use cases, and then a simple takedown of the dataset did not solve the problems because of derived datasets, alternative sources, models trained on said data, etc. The authors look at about 1000 papers, focusing on three widely used facerec datasets (MS-Celeb, Duke-MMTC and LFW) to highlight specific harms, and then provide general guidelines to reduce such harms.

---

> ### Author Response · Authors · 2021-09-26
> **Response to reviewer 83pV**
>
> Thank you for the feedback. Your comment regarding our analysis of online discussion is well taken. In our revision, which we will submit in a few days, we will more clearly state the limitations and caveats of such an analysis. For the purposes of this paper, since the discussion of licenses was just one section, we settled on this preliminary approach. We agree that more work should be done to verify our anecdotal evidence. For example, interviews with practitioners would be valuable. We will also include this as a direction for future work in our revision.

---

> > ### Comment · Reviewer_83pV · 2021-09-30
> > **Response to revision**
> >
> > Thank you for addressing this in the revised draft. I'm happy with the changes!

---

> ### Author Response · Authors · 2021-09-29
> **Posted revision**
>
> We have posted a revised draft, which includes the points mentioned in our previous comment. New text is colored in red. Again, we appreciate your feedback.

---

### Official Review · Reviewer_3hBG · 2021-09-20
**Examining distributional harms of retracted datasets**

**Rating:** 10
**Confidence:** 4
**Correctness:** The paper seemed as correct as possible.

**Strengths:**

The main strength of this paper is to cooperate with the algorithmic auditors who have been instrumental in influencing unethical dataset retraction and demonstrate that sometimes retraction is not enough when the dataset has already been distributed widely.

The discussion of licenses is very important, and it is great that the authors tied in the legality of the situation, giving us a multidisciplinary perspective.

I 1000% agree with the authors that conferences and journals should have protocols for papers that use datasets that were retracted, and doing so is probably the single best way to prevent continued academic (broadly construed, e.g., industry research) use of the dataset.

I'm blown away by this paper, and the level of attention the authors paid to the details of this very specific domain and its nuances.

After reading the paper multiple times, I'm convinced a) this is a top NeurIPS paper and b) every person attending NeurIPS this year should read this paper. It is that important given the importance of data in machine learning and what is at stake for marginalized communities.

**Weaknesses:**

I always read with skepticism. On my first read, I was still skeptical. However, after reading a few more times, I am thoroughly convinced that this paper is a huge contribution to the literary canon.

**Additional Feedback:**

I loved the use of tables in the paper. The tables were very detailed and relevant.

**Clarity:**

The paper is rather clear and written in an appropriate sequence for the reader.

**Documentation:**

The authors provide sufficient detail on data collection and organization. The dataset is available and hosted on Github. Since it was only used for this paper, it might not be maintained, so there may eventually be broken links. The authors can correct my inference of their maintenance plan if I am wrong. Ironically, the Github does not have a license although the paper discusses licenses in detail. The authors should explicitly put a license on Github.

**Ethics:**

The ethics seem good to go.

**Relation To Prior Work:**

This paper is heavily related to prior works on algorithmic audits which have influenced dataset retraction.

**Summary And Contributions:**

In "Mitigating dataset harms requires stewardship: Lessons from 1000 papers", the authors assert although problematic datasets (e.g., biased computer vision datasets) may be retracted, they can still cause harm when they are no longer available from the original creator. The authors support this argument by highlighting that derivative datasets can still persist after retraction and unclear licenses can create ambiguity that allows ML practitioners to operate in an ambiguous legal space where they can continue to use the original and derivative dataset without repercussions. The authors' intent is to demonstrate that dataset retraction is not enough to mitigate dataset harm and that data stewardship is paramount throughout a dataset's life cycle, and to this end, the author's provide important recommendations for dataset stewardship on an individual and community level. The authors' significant contribution is that it demonstrates what happens after datasets are retracted, and I believe this paper will be heavily cited in the literature because it proves what a lot of us already knew.

---

> ### Author Response · Authors · 2021-09-26
> **Response to reviewer 3hBG**
>
> Thank you for the feedback. In our revision, which we will submit in a few days, we plan on including more details about our maintenance plan for the dataset. You are right that the links to PDFs may stop working, but we expect that the remaining columns will remain stable (in particular, DOIs will allow the papers to remain accessible, although less convenient than a direct link). You are absolutely right that we should add a license to our dataset, and we will choose an appropriate license shortly.

---

> ### Author Response · Authors · 2021-09-29
> **Posted revision**
>
> We have posted a revised draft reflecting the points we discussed in our previous comment. New text is colored in red. We have added details about maintenance in the Appendix, as well as preliminary text discussing a license. We are currently discussing the details of a license / terms of use. Again, we appreciate your feedback.

---

### Official Review · Reviewer_EUQo · 2021-09-21
**Critical research in the area of Ethics**

**Rating:** 10
**Confidence:** 5
**Clarity:** Yes, the paper clearly states the pro…

**Strengths:**

I think the paper discusses a fundamental problem with dataset usage where ethical concerns are ignored leading to the proliferation of datasets with ethical concerns. This lack of data validation then make its way to larger high scale industry systems leading to severe downstream impact. The authors have covered an exhaustive list of issues related to the use of datasets that have been identified to have serious ethical concerns. It points out limitations of widely used and accepted data management and license mechanism, which calls for significant improvements.

**Weaknesses:**

I would have also liked to see some potential mitigation strategies for ethical issues behind dataset usage.

**Additional Feedback:**

This paper highlights a very important and fundamental problem with the current state of machine learning research.

**Correctness:**

I think the claims made in the paper are correct and supported by several data points mentioned in the paper.

**Documentation:**

Yes, the documentation is sufficient.

**Ethics:**

No ethical concerns.

**Relation To Prior Work:**

This paper seems to be the first in its kind and doesnt provide  significant details of related work.

**Summary And Contributions:**

The authors present the ethical impacts of three datasets, i.e DukeMTMC, MS-Celeb-1M, and Labeled Faces in the Wild (LFW), which were retracted due to ethical concerns but continue to be used and cited by several other research papers. The authors studied around 1000 papers that cited the retracted datasets highlighting the ethical repercussions of continued usage of abjured datasets without any specified mechanism. Specifically, the paper presents analysis on five points, i.e, (1) data retraction does little to mitigate ethical issues, (2) derivatives of retracted datasets spawn new ethical concerns, (3) pitfall of data liscences, (4) changes in ethical concerns over time and its impact, and (5) limitation of data management and citation practices which do not seem to pacify ethical concerns.

---

> ### Author Response · Authors · 2021-09-26
> **Response to reviewer EUQo**
>
> Thank you for the feedback. We view our work as complementary to the existing literature on mitigation strategies, which we cite in the first paragraph on Section 8. We plan on submitting a revised version in a few days, and we will elaborate this paragraph to better describe and summarize existing ideas. We will also expand some of our recommendations for stakeholders in section 8. If there is a specific component pertaining to dataset usage where you would like to see more discussion, please let us know.

---

> ### Author Response · Authors · 2021-09-29
> **Posted revision**
>
> We have posted a revised draft reflecting the points we discussed in our previous comment. In particular, we have elaborated on existing mitigation strategies, and added additional detail to our own recommendations. New text is colored in red. Again, we appreciate your feedback.

---

### Official Review · Reviewer_SBkR · 2021-09-22
**A helpful critical guide to dataset practices with implications for best practices**

**Rating:** 8
**Confidence:** 4
**Correctness:** Yes, with the caveats presented.
**Clarity:** Yes

**Strengths:**

This paper would be welcome at any time, but it is especially fitting that it accompany the first iterations of the NeurIPS benchmark track.  As the benchmark track finds its feet and continues to create and revise community norms regarding the creation and maintenance of benchmarks, a paper that combines a survey of current practices with a normative analysis of where they fall short seems of great relevance. The contribution is significant, as it draws together existing descriptive and normative work while addressing a novel topic. Overall I consider the paper to be an important contribution to the benchmarks track and worthy of acceptance.

**Weaknesses:**

Specific suggestions for improvement noted in "additional feedback" below.  My overall concern with the paper was with its normative analysis. It followed a common practice in AI ethics of analyzing ethics primarily in terms of possible harms/risks of harms, but more clarity was needed in attributing responsibility for harms or risks mentioned.

**Additional Feedback:**


Section 3: It would be worth citing here the literature on citations of retracted scientific papers, which shows a similar effect.  For example, Brainard, Jeffrey, and Jia You. "What a massive database of retracted papers reveals about science publishing’s ‘death penalty’." Science 25.1 (2018): 1-5.
A comparison & contrast with retraction of scientific papers might further illuminate the possible steps forward. In particular it seems worth nothing that in scientific paper retraction, a reason typically is given and the paper remains archived, two points raised in Section 3, but unfortunately the paper itself often remains available without its retraction context, as described in:
da Silva, Jaime A. Teixeira, and Helmar Bornemann-Cimenti. "Why do some retracted papers continue to be cited?." Scientometrics 110.1 (2017): 365-370.
One might worry that if the retraction reason is not archived in every place the dataset is archived, the existence of a publicly-accessible reason will not do as much to prevent continued use as might be hoped.

Section 4:

I was not clear on how to interpret the points presented in this section — what should be done to mitigate this kind of harm or who was responsible for the harm.  It seems very true that new applications may raise new ethical concerns, and that people who are not the dataset’s author may transform the datasets in ways that make previously benign datasets problematic, but what is the normative significance of this? Is the implication that this is bad because the authors may be associated with the new applications (and therefore in some way responsible for them?) but have no control over their production? Or is the concern that transformative use of the dataset is in some way different in kind than the same bad or misguided actors creating their own irresponsible dataset from scratch?  If this is an ethical risk, is it one that dataset authors can reasonably predict or control? My sense is that it is not — there is nothing a responsible dataset author could do to prevent later irresponsible use of a dataset — so I am not sure what stewardship could mitigate any of the dataset harms mentioned here.

Section 6:
I thought the story in this section was much more interesting and nuanced than the summary at the beginning of the first paragraph or the quotation at the end of the last paragraph.  What is interesting about the cases you raise is that they are not the same. In the case of LFW, auditing and activist work in combination with increased public awareness and concern about algorithmic bias/fairness seems to have brought about a genuine shift in analysis of the “diversity” of the dataset. By contrast the ImageNet story, as the paper notes on 216, is murkier in that the offensive labels would have been similarly offensive when ImageNet was still released.  Thus the story can’t be summarized as “what society deems fair and ethical changes over time.”

Section 7
Harms and misuses related to current practices were especially clear & well connected here.

Line 343-346 does not seem actionable.

**Documentation:**

N/A

**Relation To Prior Work:**

Yes

**Summary And Contributions:**

The paper analyzes the lives and afterlives of three popular datasets, two of which have been retracted, in a corpus of papers that cite them. It finds:
(1) that retracted datasets still circulate and that reasons for their removal are not always clearly stated or archived in order to normatively discourage continued use;
(2) that derivatives of datasets, some with additional labels, introduce additional ethical concerns, namely new applications, use of the dataset to create pre-trained models, new annotations some of which are problematic, and post processing which may introduce biases.
(3) that dataset licenses requiring non-commercial use do not stop
(4) that critique of datasets may lag behind their release on the order of years, due in part to technological or social/cultural change;
(5) that citational practices and infrastructure could be improved in order to better support tracking of dataset use and misuse.

---

> ### Author Response · Authors · 2021-09-26
> **Response to reviewer SBkR**
>
> Thank you for the feedback. We plan on submitting a revised version in a few days, but wanted to take this chance to outline our plan in case you would like to respond.
>
> Section 3: Thank you for the relevant examples. We think a brief discussion of issues related to the retraction of scientific papers will provide valuable context. In particular, the point about papers being available without their retraction context suggests that a clear and public notice of retraction may not be sufficient. We will add such discussion to our next revision.
>
> Section 4: We agree that this section could use more discussion. To address your questions, we propose adding two clarifying points in a revision. First, we agree that a derivative dataset that is ethically problematic is no more harmful than that same dataset being created from scratch. However, the effort or cost of creating a derivative can be much smaller, by several orders of magnitude in some cases, than creating it from scratch (we will give examples in our revision). Further, we agree that dataset authors cannot always anticipate the creation of ethically problematic derivatives. While we do not blame dataset authors for harmful downstream use, including derivatives, we nonetheless advocate for allocating partial responsibility to them. This is because we do think there are steps that can decrease the risk of ethically problematic derivatives. Dataset authors could specify acceptable and unacceptable types of derivatives in the dataset’s Terms of Use. Of course, there will be a grey area between derivatives known in advance to be acceptable and those known in advance to be unacceptable. Authors could consider adding a clause to the Terms of Use that requires derivative creators to obtain permission unless the derivative belongs to a pre-approved category. We will elaborate these points and add them to our revision in Section 4 and Section 8.1 (Recommendations for dataset creators).
>
> Section 5: Thank you for these observations. We plan on reworking the introduction and conclusion of this section to reflect and highlight the nuances you point out.

---

> > ### Comment · Reviewer_SBkR · 2021-09-27
> > **Affirmation of Plan**
> >
> > Dear authors,
> >
> > The plan you outline sounds good to me and responds to all of the comments -- looking forward to reading the revision.
> >
> > Best, reviewer SBkR

---

> > > ### Author Response · Authors · 2021-09-29
> > > **Posted revision**
> > >
> > > We have posted a revised draft reflecting the points we discussed in our previous comment. New text is colored in red. Again, we appreciate your feedback.

---

> > > > ### Comment · Reviewer_SBkR · 2021-09-30
> > > > **Thank you!**
> > > >
> > > > Thanks to the authors for their thoughtful responses. They have fully addressed my concerns -- I found the clarification of section 4 particularly helpful. The changes add to an already very strong paper and make it a clear candidate for acceptance, in my view.

---

### Decision · Program_Chairs · 2021-10-09

**Decision:**

Accept

**Comment:**

There is strong consensus amongst the reviewer that this paper offers a valuable contribution to this track. Reviewers are in agreement that the contribution is significant and timely and especially appropriate for the first iteration of this new NeurIPS track. I am very pleased to recommend acceptance into this track as an oral presentation.